# Rescue of *Iqsec2* Knockout Mice with Human IQSEC2 Adeno-Associated Virus Mediated Gene Therapy

**DOI:** 10.3390/ijms26178311

**Published:** 2025-08-27

**Authors:** Divyalakshmi Soundararajan, Emi Kouyama-Suzuki, Yoshinori Shirai, Shaun Orth, Veronika Borisov, Yonat Israel, Yisrael Weiss, Leah Avi-Isaac, Niguse H. Garoma, Orit Lache, Nina S. Levy, Suyao Li, Weichen Zang, Shai Netser, Shlomo Wagner, Gabriel Jimenez, Wayne N. Frankel, Katsuhiko Tabuchi, Tristan T. Sands, Andrew P. Levy

**Affiliations:** 1Center for Translational Research in Neurodevelopmental Disease, Department of Neurology, Columbia University Vagelos College of Physicians and Surgeons, New York, NY 10032, USA; ds4013@cumc.columbia.edu (D.S.); gj2373@cumc.columbia.edu (G.J.); wf2218@cumc.columbia.edu (W.N.F.); tristan.sands@columbia.edu (T.T.S.); 2Department of Molecular and Cellular Physiology, Shinshu University School of Medicine, Matsumoto 390-8621, Japan; emi_suzuki@shinshu-u.ac.jp (E.K.-S.); yoshirai@shinshu-u.ac.jp (Y.S.); 24hm136b@shinshu-u.ac.jp (S.L.); 25hm114e@shinshu-u.ac.jp (W.Z.); ktabuchi@shinshu-u.ac.jp (K.T.); 3Technion Faculty of Medicine, Technion Israel Institute of Technology, Haifa 3109601, Israel; shaun.orth@campus.technion.ac.il (S.O.); veronikabor@campus.technion.ac.il (V.B.); israel.yonat@campus.technion.ac.il (Y.I.); amyisraelchai18@gmail.com (Y.W.); lyh@campus.technion.ac.il (L.A.-I.); nigusegaroma@campus.technion.ac.il (N.H.G.); eorit@technion.ac.il (O.L.); ninal@technion.ac.il (N.S.L.); 4Sagol Department of Neurobiology, Faculty of Natural Sciences, University of Haifa, Haifa 3103301, Israel; snetser@univ.haifa.ac.il (S.N.); shlomow@research.haifa.ac.il (S.W.)

**Keywords:** IQSEC2, adeno-associated virus, gene therapy, epilepsy, intellectual disability, autism

## Abstract

The IQSEC2 protein is a guanine nucleotide exchange factor for Arf6. Pathogenic variants in the X-linked *IQSEC2* gene are associated with drug-resistant epilepsy, severe intellectual disability, and autism. The vast majority of disease-causing variants introduce premature termination codons into the *IQSEC2* gene, resulting in little or no IQSEC2 protein being produced. Approximately 20% of cases are missense variants in the seven functional domains of the IQSEC2 protein. We sought to determine whether an adeno-associated virus (AAV) containing the *IQSEC2* gene could rescue abnormal phenotypes in mice in two different *Iqsec2* mouse models with premature *Iqsec2* termination codons resulting in a knockout of the *Iqsec2* gene expression and in mice with an A350V *Iqsec2* missense mutation. In the *Iqsec2* knockout mice, the AAV significantly improved growth, corrected behavioral abnormalities, and normalized the seizure threshold. Behavioral abnormalities were partially rescued in A350V mice, which expression studies suggest may have been due to the feedback inhibition of the endogenous *Iqsec2* allele by viral *IQSEC2*. We propose that the success in the *Iqsec2* knockout mice warrants a proof-of-concept study for gene replacement therapy in boys with *IQSEC2* premature termination variants.

## 1. Introduction

The *X*-linked gene *IQSEC2* encodes a protein of 1488 amino acids found at the post-synaptic density (PSD) of neurons throughout the brain [1]. IQSEC2 functions as a guanine nucleotide exchange factor (GEF) for Arf6, promoting the exchange of GDP for GTP on Arf6 and thereby activating Arf6 [2]. Activated Arf6-GTP, in turn, mediates AMPA receptor trafficking, actin polymerization, and dendritic spine maturation [3,4,5]. As such, the IQSEC2 protein plays a major role in controlling synaptic transmission, the excitatory–inhibitory balance, and memory consolidation.

Shoubridge and colleagues first reported in 2010 on four affected individuals with *IQSEC2* variants and severe intellectual disability, autism spectrum disorder, and drug-resistant seizures [6]. Today, over 1000 cases and 150 different mutations have been documented in the *IQSEC2* gene associated with disease [1]. Children present as early as 6 months of age with hundreds of brief daily seizures, with the vast majority of affected males completely unresponsive to antiseizure medication; the few responders require as many as five–six medications to attain only partial control. Children with *IQSEC2*-associated disease are minimally verbal or nonverbal, have an IQ below 50, and will need constant lifetime caregiver supervision [1].

Approximately 80% of all *IQSEC2* disease-causing variants are due to the generation of premature termination codons in exons 1–14 of the *IQSEC2* gene [1,7], which are predicted to result in no IQSEC2 protein being produced due to nonsense-mediated RNA decay (NMD) [8]. In addition, there are disease-causing missense variants in the regions of the *IQSEC2* gene encoding the seven functional domains of the IQSEC2 protein, with some of these representing a partial loss of function and others representing an increase in or gain of function [9]. Accordingly, the treatment of *IQSEC2* disease may need to be personalized based on the functional genomics of the specific disease allele in any given child.

Gene therapy, using adeno-associated virus (AAV), is emerging as a means for providing clinically meaningful benefit for an increasing number of diseases caused by mutation in a single gene. However, a major issue in the use of AAV is the size limitation of the AAV genome required for the efficient packaging of the AAV. The maximum permitted size of the DNA genomic cargo for the generation of high-titer AAV is under 5.0 kb, with viral genomes greater than this size being associated with substantially lower titers and a decrease in the quality of the virus (empty viral capsids and partial deletions of the genome) [10].

Previous work has demonstrated that an AAV with a genome of over 5.1 kb containing the rat *Iqsec2* gene regulated by a non-cell- and non-tissue-specific EF1a promoter (referred to in this study as the rIQSEC2 AAV) could partially rescue behavioral abnormalities seen in an *Iqsec2* knockout mouse model [11]. In the present study, we sought to determine whether an AAV with a genome of 4.9 kb containing the human *IQSEC2* gene regulated by a small-neuron-specific promoter [12] (referred to in this study as the hIQSEC2 AAV) could rescue abnormal phenotypes in mice with an *Iqsec2* knockout mutation or in mice with a human *IQSEC2* missense paralog.

## 2. Results

In this study, we used two IQSEC2-encoding AAVs, which differ in (1) the species origin of the *IQSEC2* gene, (2) the promoter driving *IQSEC2*, and (3) the length of the AAV genome. The first virus (rIQSEC2 AAV) contains rat *Iqsec2*, has a non-cell-specific promoter, and is 5.1 kb in length. The second virus (hIQSEC2 AAV) contains human *IQSEC2*, has a neuron-specific promoter, and is 4.9 kb in length. The titer of the rIQSEC2 AAV tended to be lower than that of the hIQSEC2 AAV (see Section 4) and showed increased deleted viral genomes, as evidenced by the smearing and lower-molecular-weight bands on an agarose gel when compared to the hIQSEC2 AAV (Figure 1). These results are consistent with prior reports showing truncations in the packaging of AAV genomes of approximately 5.2 kb or greater [10].

### 2.1. The hIQSEC2 AAV Rescues Abnormal Phenotypes in Mice with a Phe860Serfs*8 (Knockout) Iqsec2 Mutation at Columbia University

The *Phe860Serfs*8 Iqsec2* mutation generates a premature termination codon eight amino acids downstream of a one-nucleotide deletion in the codon for amino acid 860. Hemizygous males have previously been demonstrated to make no detectable IQSEC2 mRNA or protein [13]. In the previously reported characterization of the Phe860Ser*fs8 *Iqsec2* knockout model, the most robust differences in the phenotypes between hemizygous male mutant mice and their wild-type littermates were those associated with impaired neonatal growth and unusual seizure resistance to 6 Hz electroconvulsive stimulation [13]. Therefore, the growth and electroconvulsive threshold (ECT) phenotypes were used to assess the efficacy of the rIQSEC2 AAV or hIQSEC2 AAV in hemizygous or heterozygous mice and their wild-type littermates.

For growth in male mice, there was a significant interaction between the *Iqsec2* genotype (wild type or knockout) and treatment (type of AAV) for both the rIQSEC2 AAV (repeated-measures MANOVA; *p* = 0.017) and the hIQSEC2 AAV (repeated-measures MANOVA; *p* < 0.00001) (Figure 2A). In pairwise comparisons, between postnatal day eight (PND 8), when the AAV gene expression is first detectable [14], and PND 21 (weaning age), we found that there was no change in the growth rate with the rat IQSEC2 AAV (F = 0.03, *p* = 0.86) compared to the controls in hemizygous mutant (knockout) males (Figure 2B), but there was a significant improvement in the growth in hemizygous mutant (knockout) males receiving the hIQSEC2 AAV (F = 0.149, *p* = 0.02) compared to the controls (Figure 2C). The difference in the weights of the controls vs. those of the hIQSEC2 AAV-treated hemizygous mice became less statistically significant after weaning at PND 21, reflecting the increase in the variability in the weights of the mice after weaning. In wild-type littermates during the PND 8-PND 21 interval, both the hIQSEC2 AAV and rIQSEC2 AAV resulted in a significant impairment in the growth rate (control vs. human IQSEC2, *p* = 0.007; control vs. rat IQSEC2, *p* = 0.039). In heterozygous (het) knockout females, between PND 8 and PND 21, there was a significant impairment in the growth rate with the rIQSEC2 AAV (*p* = 0.028), with no effect seen in WT female or het knockout female mice with the hIQSEC2 AAV (Appendix A).

In the ECT test as previously reported [13], hemizygous knockout males demonstrated an increased seizure threshold compared to wild-type mice. Heterozygous knockout female mice demonstrated an intermediately elevated seizure threshold between that of hemizygous IQSEC2 knockout mice and wild-type mice (Figure 3). The hIQSEC2 AAV significantly normalized the ECT threshold in the hemizygous males but the rIQSEC2 AAV did not. A similar benefit on the ECT threshold was seen in female heterozygous KO mice with the hIQSEC2 AAV (*p* = 0.056). No change in the ECT threshold was observed in wild-type mice with either AAV.

**Mortality**. At six months of age, the mortality in the control hemizygous *Iqsec2* knockout male mice was 14% (3/21 mice), compared to no deaths in the hemizygous *Iqsec2* knockout males treated with either the rIQSEC2 AAV (0/13) or hIQSEC2 AAV (0/18).

### 2.2. The hIQSEC2 AAV Partially Rescues Abnormal Phenotypes in Mice with a Ser254* (Knockout) Iqsec2 Mutation at Shinshu University

The Ser254* *Iqsec2* mutation, generated in the Tabuchi lab [11] at Shinshu University, has a 17 bp deletion in exon 3 of the mouse *Iqsec2* gene, thereby generating a premature termination codon after Ser254. Hemizygous males for this mutation were previously demonstrated to make no IQSEC2 protein [11]. In the previously reported characterization of the Ser264* *Iqsec2* knockout model at Shinshu University [11], it was demonstrated that male mice hemizygous for the knockout mutation displayed impairments in the social preference and social novelty preference tests. In the present study, similar to what was previously reported in this lab, the rIQSEC2 AAV partially rescued the social preference abnormality but did not show a significant benefit in the novelty test. However, the hIQSEC2 AAV showed a significant benefit in partially rescuing abnormalities in both the social preference and social novelty tests, as shown in Figure 4.

### 2.3. The hIQSEC2 AAV Does Not Rescue Abnormal Phenotypes in Mice with an A350V Iqsec2 Mutation at Technion

The A350V *IQSEC2* mutation is a human disease variant in which the ability of calmodulin to negatively regulate the Sec7-catalyzed GEF activity of IQSEC2 has been lost, thereby resulting in the constitutive activation of the GEF activity of IQSEC2 and the constitutive elevation of Arf6-GTP [9]. Mice with the A350V *Iqsec2* mutation have been demonstrated to show several abnormal phenotypes compared to their wild-type littermates [1]. Among these phenotypes, the most robust are lethal seizures occurring between days 15 and 20 (approximately 25% mortality during this interval) [15] and abnormal social communication (marked reduction in male vocalizations in response to a female stimulus at 2 months of age) [16]. We sought to determine whether the rIQSEC2 or hIQSEC2 AAV could rescue these phenotypes in the A350V *Iqsec2* model. We found that there was no benefit to the seizure-induced mortality by either the rIQSEC2 AAV (6/29 (21%)) or the hIQSEC2 AAV (10/25 (40%)) compared to the A350V *Iqsec2* mice receiving no treatment (25/108 (23%) (*p* = 0.77) for the rIQSEC2 AAV and *p* = 0.08 for the hIQSEC2 AAV). Neither the rIQSEC2 nor the hIQSEC2 AAV induced visible seizures or mortality in the wild-type mice. Furthermore, the hIQSEC2 AAV did not alleviate the deficit in vocalizations displayed by A350V male mice towards a female (Figure 5). Conversely, the hIQSEC2 AAV administered to male wild-type *Iqsec2* mice resulted in a significant reduction in male vocalizations (Figure 5). Likewise, for the rIQSEC2 AAV, the virus failed to alleviate the deficit in vocalizations in the majority of A350V mice. However, for the rIQSEC2 AAV, approximately 20% (5/23) of the mice demonstrated levels of vocalizations seen in untreated wild-type mice (greater than 500 USFs).


*Assessment of mouse brain IQSEC2 mRNA after treatment with the hIQSEC2 AAV supports an autoregulatory mechanism for IQSEC2 mRNA.*


We wished to determine the changes in the amount of endogenous mouse *Iqsec2* mRNA expression in the brain due to treatment with the hIQSEC2 AAV. Surprisingly, we found that in brains injected with the hIQSEC2 AAV, the endogenous mouse *Iqsec2* mRNA was decreased 2.5-fold (treated/untreated: 0.4 ± 0.1, *n* = 15) compared to wild-type untreated mice. We propose that these data demonstrate the existence of a feedback inhibition for *Iqsec2* mRNA operating on the endogenous mouse *Iqsec2* gene, as depicted in Figure 6. We suggest that the reduction in endogenous mouse *Iqsec2* mRNA after treatment with the hIQSEC2 AAV is due to a downregulation of the transcription of the endogenous mouse *Iqsec2* mRNA by the IQSEC2 protein from the hIQSEC2 AAV. The viral promoter is not subject to feedback inhibition, nor is the stability or translation of the viral mRNA subject to the regulatory mechanisms present for the endogenous mouse *Iqsec2* mRNA. Thus, even though there is less endogenous mouse *Iqsec2* mRNA after hIQSEC2 AAV treatment, there may be normal or even elevated levels of IQSEC2 protein, as the viral *IQSEC2* mRNA may have an increased translational efficiency. We were unable to identify specific primers which could differentiate rat (viral) from mouse *Iqsec2* for measuring endogenous mouse *Iqsec2* mRNA after treatment with the rIQSEC2 AAV.

The feedback inhibition mechanism may explain the apparent rescue of vocalizations in a few A350V mice after treatment with the rIQSEC2 AAV. We examined viral *Iqsec2* mRNA derived from the rIQSEC2 AAV using specific primers to detect virally encoded *Iqsec2* mRNA as opposed to endogenous mouse *Iqsec2* mRNA. We found that in those mice with the highest levels of viral *Iqsec2* mRNA, there was a complete rescue of vocalizations, with a significant correlation between the amount of rat *Iqsec2* mRNA from the virus and vocalizations (*n* = 26, r = 0.80, *p* < 0.0001) (Figure 7). We propose that in these mice with very high levels of viral IQSEC2, there was a complete downregulation of mRNA from the diseased A350V *Iqsec2* allele, producing an effective knockdown of the A350V *Iqsec2* disease allele mRNA and allowing for rescue of the vocalization phenotype.

## 3. Discussion

We have demonstrated in this study that an AAV expressing the human *IQSEC2* gene under the control of a neuron-specific promoter can rescue multiple abnormal phenotypes related to the behavior and seizure susceptibility in *Iqsec2* mice that do not produce any IQSEC2 protein as a result of a knockout of the *Iqsec2* gene. A loss of IQSEC2 protein expression, due to a premature termination codon and the consequent nonsense-mediated RNA decay (NMD) [8], is thought to occur in approximately 80% of all pathological *IQSEC2* mutations in male children [1], thus underscoring the importance of the findings presented here for the IQSEC2 community of families, clinicians, and researchers. The human IQSEC2 AAV used in this study was prepared, tested, and shown to benefit *Iqsec2* knockout mice in two independent laboratories, thereby providing increased confidence and assurance as to the robustness of the benefit of this virus in the knockout condition.

The differences in the efficacy between the two viruses used in this study (the rIQSEC2 AAV and hIQSEC2 AAV) are most likely due to differences in the IQSEC2 promoters used for the two viruses, with the rIQSEC2 AAV allowing for ectopic (non-neuronal) expression, while the hIQSEC2 AAV permits expression only in neurons. We have also shown that the larger genome of the rIQSEC2 AAV (5.1 kb) as compared to the hIQSEC2 AAV (4.9 kb) results in a slightly lower viral titer with partial degradation of the viral DNA, as has been reported for other AAVs with genomes of approximately 5.2 kb or greater [10]. Sequence differences between the two viral constructs were minimal and not likely to account for result disparities.

We propose a novel feedback inhibition mechanism regulating steady-state levels of mRNA. The evidence for this hypothesis is based on the observation that endogenous (mouse) *Iqsec2* mRNA is decreased with the hIQSEC2 AAV. We suggest that the IQSEC2 protein made by the virus serves to downregulate the amount of endogenous IQSEC2 transcription from the mouse *Iqsec2* gene. This feedback inhibition could explain how the amounts of IQSEC2 protein are similar in males and females despite the fact that the human *IQSEC2* gene escapes X inactivation [1]. We have also proposed that this feedback inhibition mechanism may explain the apparent rescue of the vocalization phenotype in A350V mice expressing very high levels of viral IQSEC2 mRNA by suppressing the production of the A350V disease allele. This rescue may have only been seen with the rIQSEC2 AAV because the E1fa promoter in the rIQSEC2 AAV is a much stronger promoter than the CALM promoter (12) present in the hIQSEC2 AAV. A higher level of IQSEC2 protein may be needed to achieve a complete knockdown of the endogenous mouse *Iqsec2* mRNA. Definitive support for the feedback inhibition mechanism will require assessment of the endogenous and viral IQSEC2 protein expression. While selective quantitation of the viral IQSEC2 protein may be achieved by encoding an N-terminal FLAG tag in the viral *IQSEC2* gene, the measurement of mouse endogenous IQSEC2 protein by immunostaining will be more challenging due to the lack of antibodies that can distinguish mouse IQSEC2 from rat and human IQSEC2. One possible solution might be to first deplete the Flag-tagged viral IQSEC2 protein from the extract by immunoprecipitation and then assess the remaining endogenous IQSEC2 in the extract by Western blot analysis.

We believe suppressing mRNA production from the A350V disease allele in combination with replacement with a functional copy of the *IQSEC2* gene can rescue A350V-associated disease. This approach could use a shRNA targeting the endogenous *IQSEC2* mRNA [11] while sparing the hIQSEC2 AAV mRNA due to a change in the codon usage in the region targeted by the shRNA. Combining the shRNA for knockdown with a replacement *IQSEC2* gene in a single AAV [17] may require identification of a small region of the *IQSEC2* open reading frame, which can be deleted without compromising the function of IQSEC2 [18].

An alternative approach to the knockdown and replacement strategy for gain-of-function mutations like A350V could be to use CRISPR-CAS9. Numerous clinical trials are underway with CRISPR to correct single-gene defects, particularly for blood and liver disorders where delivery of CRISPR is not limiting. However, currently, there are no approved therapies (unlike AAV) or clinical trials recruiting or ongoing for central nervous system disorders due to the challenges involved in central nervous system disorders, as has been summarized recently [19]. For example, classic CRISPR relies on the cells’ DNA repair pathway, with homology-directed repair being required for more precise editing; however, this type of editing is limited in post-mitotic neurons. Nonetheless, recent studies involving modifications to the CAS9 protein may overcome the limitations of classic CRISPR in the future.

We found, in two different laboratories, that the human IQSEC2 AAV was harmful to wild-type *Iqsec2* male mice, as shown by the decrease in growth and decrease in female-induced vocalizations. While the therapeutic dosage window for IQSEC2 is not known, the harmful effect of the AAV in wild-type mice may be due to overexpression of IQSEC2 [5,20]. There is precedent from other gene therapy studies that too much of the replacement gene product can cause toxicity. A prime example of this has been demonstrated with an AAV expressing the *MECP2* gene designed to treat Rett Syndrome [21], where too much of the MECP2 protein delivered by the AAV resulted in neurotoxicity. The approach taken for Rett syndrome to overcome this problem, termed EXACT technology [21], is to incorporate into the viral genome the miRNA-binding sites used by the endogenous *MECP2* gene to control the viral gene expression in each cell. The virally encoded IQSEC2 mRNA lacks the translational regulation from miRNA networks that may control IQSEC2 protein production from a given level of *IQSEC2* mRNA. We have identified three miRNA-binding sites in the human *IQSEC2* mRNA 3′UTR that are evolutionarily conserved, and we are currently evaluating the effects of incorporating these sites into the hIQSEC2 AAV genome with the goal of delivering replacement IQSEC2 protein at physiological levels. As incorporation of 3′UTR sequences into the hIQSEC2 AAV genome would increase the size of the genome, resulting in a reduction in the AAV titer and quality, we are assessing whether we can harness the miRNA regulatory pathway by introducing the *IQSEC2* miRNA-binding sites into the *IQSEC2* coding sequence. This may be achieved by altering the codon usage, allowing for the creation of the same miRNA-binding sequences normally found within the 3′UTR of *IQSEC2* without changing the amino acid sequence of the IQSEC2 protein.

Do these studies provide any insight into the window of time in which IQSEC2 AAV gene therapy must be administered in order to have benefit? It is encouraging that the group from Shinshu was able to demonstrate the rescue of behavioral abnormalities in adult mice. It has been suggested that gene therapy for genes involved in the synaptic structure may be more forgiving in terms of the developmental window due to synaptic plasticity [22].

### Limitations

There are limitations in the extension and translation of these results. We did not assess the in situ expression of IQSEC2 protein after viral treatment. This limits our ability to know whether the heterogeneity between the mice could be due to differences in the biodistribution of the virus between the mice. Furthermore, while we demonstrated the replication of the benefit of the hIQSEC2 AAV in independent labs in the knockout model, the assays used to assess efficacy were not the same at both sites, so this does not represent a true biological replication. Finally, we have shown the benefit of the IQSEC2 AAV in mice which make no *Iqsec2* mRNA or protein, and while it is likely that NMD results in very little if any IQSEC2 protein being produced for most of the *IQSEC2* premature termination codon mutations present in children, some human premature termination codons may have a partial escape from NMD [23].

## 4. Materials and Methods

### 4.1. Mouse Iqsec2 Mutant Models Used in This Study

Knockout model #1 (*Phe860Serfs*8 Iqsec2*). Previously described knockout of the *Iqsec2* gene generated for the Frankel lab [13] at Columbia University due to a premature termination codon 8 amino acids downstream of a one-nucleotide deletion in the codon for amino acid 860 (the mice have since been archived at The Jackson Laboratory, stock #035784). Hemizygous males were demonstrated to make no detectable *Iqsec2* mRNA or protein. Hemizygous males and heterozygous females were used in this prior study [13]. The mice were maintained on a C3HeB/FeJ: C57BL/6NJ hybrid background at Columbia University (New York, NY, USA) in accordance with the regulations for animal experimentation of the university.

Knockout model #2 (*Ser254* Iqsec2*). Previously described knockout of the *Iqsec2* gene generated in the Tabuchi lab [11] at Shinshu University due to a premature termination codon after Ser254 (exon 3) of the mouse *Iqsec2* gene. Hemizygous males only were used for these studies. Hemizygous males were previously demonstrated to make no IQSEC2 protein. The mice were maintained on a C57BL6/JJcl:129+Ter/SvJcl hybrid background at Shinshu University (Matsumoto, Japan) in accordance with the regulations for animal experimentation of the university.

IQSEC2 gain-of-function model (*A350V Iqsec2*). Previously described missense mutation substituting alanine for valine at amino acid 350 in the IQ domain of *Iqsec2* [9]. This mutation results in the constitutive activation of the GEF activity of IQSEC2. Hemizygous males only were used for these studies. The cortical IQSEC2 mRNA and protein levels in these mice have been demonstrated by qRT-PCR and Western blot to be similar to those of wild-type mice. The mice were maintained on a C57Bl/6J background at the Technion Israel Institute of Technology Medical School animal facility (Haifa, Israel) in accordance with the regulations for animal experimentation of the university.

### 4.2. AAVs Used in This Study

Rat *Iqsec2* (rIQSEC2). Previously described as AM004 [11], this AAV contains an AAV ITR-to-ITR genome of 5.113 kb with an EF1alpha short form (EFS) promoter (210 bp) driving the expression of a Flag-tagged rat IQSEC2 (1488-amino acid isoform using coding sequence of rat IQSEC2 from NM_001277425.2).

Human *IQSEC2* (hIQSEC2). This AAV contains an AAV ITR-to-ITR genome of 4.938 kb with a mini-calmodulin 1 gene promoter (120 bp) [12] driving the expression of human IQSEC2 (1488-amino acid isoform using coding sequence of human IQSEC2 from NM_001111125.3).

### 4.3. Generation and Injection of Virus

#### 4.3.1. Shinshu University

AAV was generated using the triple-transfection method [11]. Fifty to seventy percent confluent AAVpro 293T cells grown in DMEM supplemented with 10% FBS were transfected with pHelper plasmid (20 mg/dish), pRC-DJ plasmid (11 mg/dish), and each pAAV plasmid (10 mg/dish) using acidified PEI in 5 15-cm dishes. Two days after transfection, cells were collected and centrifuged at 1000 rpm for 5 min. The supernatant was removed, and the cells were resuspended in PBS and subjected to a freeze–thaw cycle 3 times. Following the addition of Benzonase (Sigma-Aldrich, Kanagawa, Japan), the cell lysate was incubated at 37 °C for 45 min and centrifuged at 3600 rpm for 15 min. The supernatant was further purified by discontinuous iodixanol gradient ultracentrifugation using Optima XE-90 Ultracentrifuge (Beckman Coulter, Tokyo, Japan). Titers are expressed as viral genomes per mL (vg/mL). The titers for the AAVs were 3.3 × 10^11^ (vg/mL) (pAAV-hIQSEC2); 3.43 × 10^11^ (vg/mL) (pAAV-rIQSEC2); 3.13 × 10^11^ (vg/mL) (pAAV-EFS-EGFP); and 3.63 × 10^11^ (vg/mL) (pAAV-H1/U6-hSyn-Tag-RFP).

Injection of the AAV was performed stereotactically. *Iqsec2* knockout or their WT littermate mice at 8–16 weeks of age were anesthetized with an intraperitoneal injection of an anesthesia cocktail (3% Dexmedetomidine Hydrochloride + 8% Midazolam and 10% Butorphanol Tartrate by volume in saline; 0.1 mL/10gm body weight of mouse). Approximately 300 nL of AAV: 3:1 by volume of Rescue Virus (AAV-EFS-rIQSEC2 or AAV-Calm-hIQSEC2): AAV-H1/U6-hSyn-Tag-RFP virus or 3:1 by volume of control virus (AAV-EFS-EGFP): AAV-H1-hSyn-EGFP virus) was injected bilaterally in the stereotactic co-ordinates of the mPFC: anterioposterior (AP) + 1.9 mm, mediolateral (ML) ± 0.2 mm, and dorsoventral (DV)—1.5 mm taking bregma as the reference point using a stereotaxic device (Narishige, Tokyo, Japan). Two weeks after AAV injection, the animals were used for experiments.

#### 4.3.2. Columbia University

AAV was generated at the Horae Gene Therapy Center of the UMass Chan Medical School in Worster, MA, USA, using the triple-transfection method using a pHelper plasmid, an AAV9 capsid plasmid, and an *Iqsec2* plasmid AM004 (rat IQSEC2) or 40 (human *IQSEC2*). The titer of AAV using the rat IQSEC2 AAV was 1.1 × 10^13^ vg/mL, and for the human IQSEC2 AAV it was 1.3 × 10^13^ vg/mL. Virus was injected at PND 1 intraventricularly (10 μL per mouse) free-hand using a 10 μL Hamilton Neuros Syringe (catalog #65460-06); Hamilton Company, Reno, NV, USA at approximately two-fifths the distance from the lambda suture to each eye. Control mice were injected with either saline or not injected at all, with no difference in the parameters measured in this study between mice receiving saline or no treatment.

#### 4.3.3. Technion

AAV was generated at the ECLV viral core at Hebrew University (Jerusalem, Israel) using the triple-transfection method (pHelper, an AAV9 capsid plasmid, and an IQSEC2 plasmid (human *IQSEC2*)). The titer obtained for rIQSEC2 was 2 × 10^12^ vg/mL, and for the hIQSEC2 AAV it was 8 × 10^12^ vg/mL. Virus was injected at PND1 intraventricularly (4 μL per mouse) free-hand using a 10 μL Hamilton Syringe (catalog #7635-01) at approximately two-fifths the distance from the lambda suture to each eye.

### 4.4. Assessment of Mouse Behavior and Seizure Susceptibility

#### 4.4.1. Shinshu University

##### Three-Chamber Social Behavior Tests

Three-chamber tests were performed as described previously [11]. For behavioral tests, mice were exposed to a 10 min habituation phase in an empty plastic cage without bedding. The apparatus consisted of a rectangular three-chamber box (each chamber was 20 × 40 × 25 cm) made of clear Plexiglas, with small openings (5 × 3 cm) between chambers that allowed free movement. After a 10 min habituation period during which the subject mouse was allowed to explore all three chambers with empty wire cups placed in the side chambers, the sociability test was conducted.

In the social preference test, a novel male mouse (stranger 1, S1) was enclosed in a wire cup and placed in one of the side chambers, while the other side chamber remained empty. The subject mouse was placed in the central chamber and allowed to explore all chambers freely for 10 min. The time spent in the vicinity (within 2 cm) of each cup was recorded using a video camera and analyzed by a trained observer blinded to the genotype and treatment. The social preference index was calculated as follows:Social Preference Index = Time spent around S1 − Time spent around E

In the social novelty preference test, the previously empty cup was now occupied by a second unfamiliar male mouse (stranger 2, S2), while the original S1 mouse remained in place. The subject mouse was again allowed to explore for 10 min. The social novelty preference index was calculated as follows:Social Novelty Preference Index = Time spent around S2 − Time spent around S1

A positive preference index indicates a preference for the social stimulus (S1 or S2) over the control (E or S1). The positions of S1 and S2 were counterbalanced across trials to avoid location bias. Behavioral experiments were performed during the light phase (9:00–18:00), and all data were collected and analyzed under blinded conditions.

#### 4.4.2. Columbia University

##### Growth and Survival Monitoring

Between PND 3 and 30, mice were weighed approximately 2–3 days per week until PND 30. Mice were weaned at PND 21. In addition, survival was monitored through 6 months of age.

##### Seizure Testing

The 6 Hz electroconvulsive test (ECT) induces focal or “psychomotor” seizures and, in particular, is considered to be useful for studying pharmacoresistant epilepsy of limbic origin. The 6 Hz electroconvulsive threshold (ECT) test was as described previously with minor modifications [24] using a model 7801 electroconvulsive device (Ugo Basile, Gemonio, Italy). The ECT test was performed on adult mice 6–12 weeks of age once daily with the stimulus intensity incrementally adjusted until the minimal current sufficient to elicit an observable focal seizure was observed. Group means were calculated to determine the threshold and effect of the virus treatment.

##### Technion

Assessment of ultrasonic vocalizations in socially interacting mice

Ultrasonic vocalizations were recorded using a condenser ultrasound microphone (Polaroid/CMPA, Avisoft Bioacoustics, Glienicke/Nordbahn, Germany). The microphone was connected to an ultrasound recording interface (UltrasoundGate 116Hme, Avisoft Bioacoustics, Glienicke/Nordbahn, Germany), which was plugged into a computer equipped with the recording software Avisoft Recorder USGH version 4.2.29 (sampling frequency: 250 kHz; FFT-length: 1024 points; 16-bit format). Ultrasonic vocalizations (USVs) were recorded in 8–10-week-old male wild-type or A350V mice during a 5 min interaction with a female C57BL/6 stimulus, following 15 min of habituation to the arena [16]. Ultrasonic vocalizations were analyzed as previously described [16] using our TrackUSF custom-made software version 1.0 (Wagner lab, Haifa, Israel) [25].

Quantitation of endogenous mouse *Iqsec2* and viral *IQSEC2* RNA

RNA extraction and real-time PCR. Brains were collected into RNA*later* solution (Invitrogen, Carlsbad, CA, USA, AM7020). Total RNA was extracted using a Hybrid-R™ total RNA Isolation kit (GeneAll Biotechnology, Seoul, Republic of Korea), and DNA was removed using a Clean-Up RNA concentrator (A&A Biotechnology, Gdansk, Poland). cDNAs were obtained using the qPCR cDNA Synthesis kit (Tamar, San Diego, CA, USA, PB30.11-10), and real-time PCRs were performed with a Fast SYBR Green master mix (Applied Biosystems, Foster City, CA, USA, AB-4385612). The primers used for the real-time PCR are listed in Table 1. Mice phosphoglycerate kinase 1 (PGK1) was used as the housekeeping gene for normalization. The real-time PCR program consisted of an initial 20 s at 95 °C, and then 30 cycles as follows: 95 °C for 1 s and 60 °C for 20 s. Quantitative real-time polymerase chain reaction (qRT-PCR) was performed on a StepOnePlus Real-Time PCR system (ThermoFisher, Waltham, MA, USA, 4376600).

Primer design. All primers were designed by the NCBI primer design tool (primer-BLAST) to have similar annealing temperatures (optimal melting temperature was set to 60 °C). Efficiency of all of the primers was assessed using the standard-curve method using serial dilutions of the cDNA and plasmid target.

### 4.5. Statistical Analyses Used in This Study

For the three-chamber behavioral tests at Shinshu, statistical analysis was performed using one-way ANOVA followed by Tukey’s post hoc test (preference indices) and unpaired Student’s *t*-test (time spent around cups). * *p* < 0.05; ** *p* < 0.01; n.s., not significant.

For growth and seizure testing at Columbia, statistical analysis was performed using JMP software (version 18). For the growth curve, the data was analyzed between the different treatment and genotype groups using MANOVA repeated measures. Each treatment group was analyzed with the control group, and the *p*-value for the overall model interaction between the genotype x treatment was determined. We also performed pairwise prespecified comparisons between specific treatment groups (i.e., male hemizygous male mice receiving control vs. AAV hIQSEC2). For seizure testing, the 6 Hz data was analyzed using LSM (standard least-square means), and the overall interaction between the groups was tabulated using Dunnett’s pairwise comparisons.

For analysis of vocalizations at Technion, MATLAB 2024a was used. The non-parametric Wilcoxon rank-sum test with Bonferroni correction for multiple comparisons was used to compare the number of USV fragments in the wild-type and A350V *Iqsec2* mice with and without human IQSEC2 AAV treatment.

## 5. Conclusions

We hope that the benefit shown here of the human IQSEC2 AAV in *Iqsec2* knockout mice will stimulate a proof-of-concept clinical study in three–four male children with premature nonsense mutations in the *IQSEC2* gene. Ultimately, we envision a personalized approach to gene therapy for *IQSEC2* disease for all children, based on the pathophysiology of the *IQSEC2* mutation and the ability to rescue electrophysiological abnormalities and synaptic dysfunction seen in patient-derived neurons for a specific *IQSEC2* mutation.

## Figures and Tables

**Figure 1 ijms-26-08311-f001:**
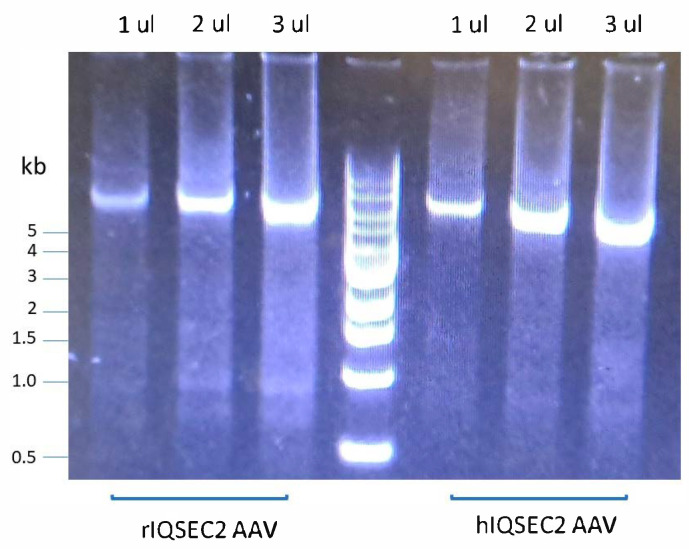
Partial degradation of rIQSEC2 AAV. A 1% agarose gel of 1–3 μL of the rIQSEC2 AAV or hIQSEC2 AAV demonstrating increased smearing and the appearance of partial breakdown products with the rIQSEC2 AAV. The titers of the rIQSEC2 AAV and hIQSEC2 AAV in these samples were 1.1 × 10^13^ viral genomes/mL and 1.3 × 10^13^ viral genomes/mL, respectively.

**Figure 2 ijms-26-08311-f002:**
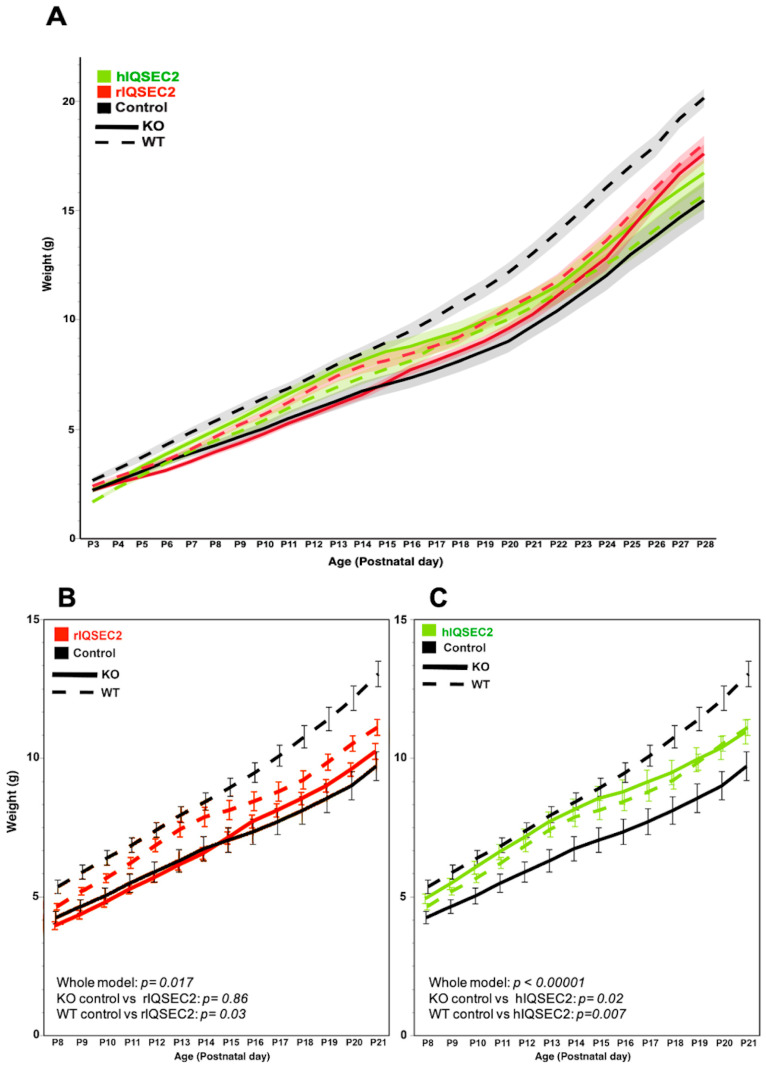
The hIQSEC2 AAV rescues the growth of neonatal male hemizygous *Iqsec2* knockout mice. (**A**) **Growth curves of male mice.** Male *Iqsec2* hemizygous knockout (KO) mice are indicated by solid lines, and male *Iqsec2* wild-type (WT) littermates are indicated by dotted lines, with the three treatment groups indicated as follows: control—black; rIQSEC2—red; human hIQSEC2—green. The shaded region for each curve is the ±SE of the mean. (**B**) **Growth curves of male hemizygous KO and male WT mice receiving rIQSEC2 AAV (red) or control (black) treatments:** MANOVA with repeated measures, *p* = 0.017 (whole model); pairwise comparisons between PND 8 and PND 21: KO (rIQSEC2 vs. control), *p* = 0.86; WT (rIQSEC2 vs. control), *p* = 0.03. Control–WT (*n* = 26), control–KO (*n* = 21); rIQSEC2-WT (*n* = 14); rIQSEC2-KO (*n* = 13). (**C**) **Growth curves of male hemizygous KO and male WT mice receiving hIQSEC2 AAV (green) or control (black) treatments:** MANOVA with repeated measures, *p* < 0.00001 (whole model); pairwise comparisons between PND 8 and PND 21: KO (hIQSEC2 vs. control), *p* = 0.02; WT (hIQSEC2 vs. control), *p* = 0.007. Control–WT (*n* = 26), control–KO (n = 21); hIQSEC2-WT (*n* = 15); hIQSEC2-KO (*n* = 18).

**Figure 3 ijms-26-08311-f003:**
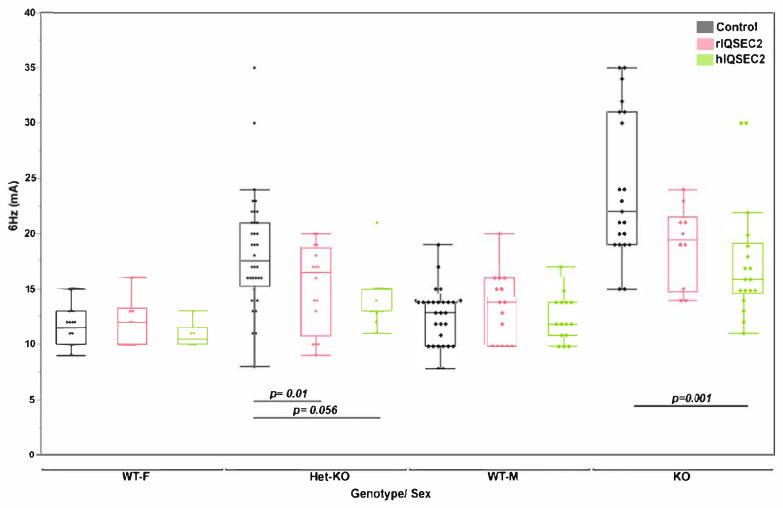
Hemizygous *Iqsec2* KO male mice demonstrate reduced 6 Hz seizure susceptibility with hIQSEC2 AAV. The seizure threshold was significantly different between the control male hemizygous mice (*n* = 21) and control male WT mice (*n* = 29) (*p* = 0.0076). The hIQSEC2 AAV significantly reduced the 6 Hz seizure susceptibility in male hemizygous KO mice (24.23 + 1.42 mA, *n* = 21, and 17.5 + 1.247 mA, *n* = 18, for control and treated mice, respectively, *p* = 0.001); no significant benefit was demonstrated in the male hemizygous KO mice with rIQSEC2 (*n* = 21 and *n* = 10 for untreated and treated KO mice, respectively, *p* = 0.57). Error bars indicate ±SEM. Similar findings were demonstrated in females.

**Figure 4 ijms-26-08311-f004:**
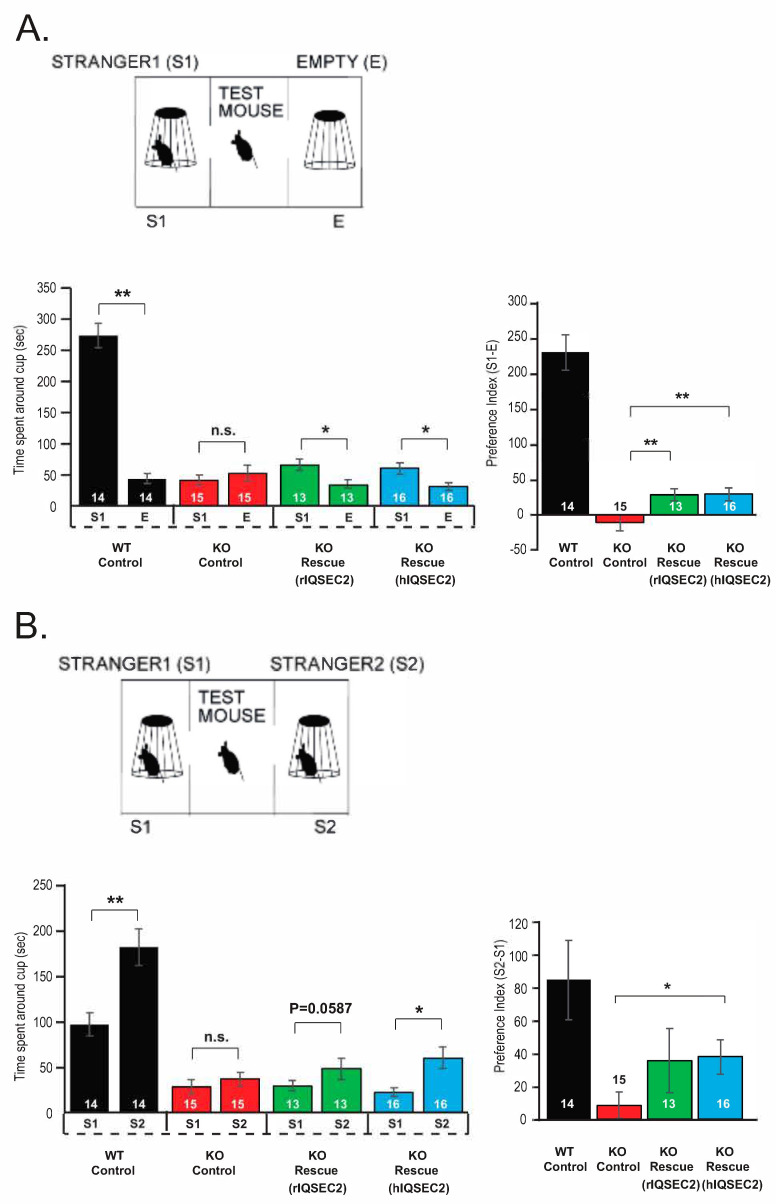
Partial rescue of social behavior deficits in *Iqsec2* knockout mice by rIQSEC2 and hIQSEC2 AAVs. (**A**) Schematic of the social preference test (top) and summary graphs (bottom) showing time spent around cups and the preference index. Four groups were tested: wild-type mice injected with the control AAV (WT Control), *Iqsec2* knockout mice injected with the control AAV (KO Control), and *Iqsec2* knockout mice injected with the rIQSEC2 AAV (KO Rescue rIQSEC2) or hIQSEC2 AAV (KO Rescue hIQSEC2). Both the rIQSEC2 and hIQSEC2 AAVs restored the preference for the stranger mouse (S1) over the empty cage (E) at similar levels, as shown by the increased time spent around the cup with the novel stimulus and positive S1–E preference index. (**B**) Schematic of the social novelty preference test (top) and summary graphs (bottom) for the time spent around cups and the preference index. The same four groups were tested for preference between a familiar mouse (S1) and a novel mouse (S2). Both the rIQSEC2 and hIQSEC2 AAVs restored the preference for the stranger mouse (S1) over the empty cage (E) at similar levels, as shown by the increased time spent around the cup with the novel stimulus and positive S1–E preference index. Data are presented as means ± SEM. Numbers of animals per group are indicated in the graphs. Statistical analysis was performed using one-way ANOVA, followed by Tukey’s post hoc test (preference indices) and unpaired Student’s *t*-test (time spent around cups). * *p* < 0.05; ** *p* < 0.01; n.s., not significant.

**Figure 5 ijms-26-08311-f005:**
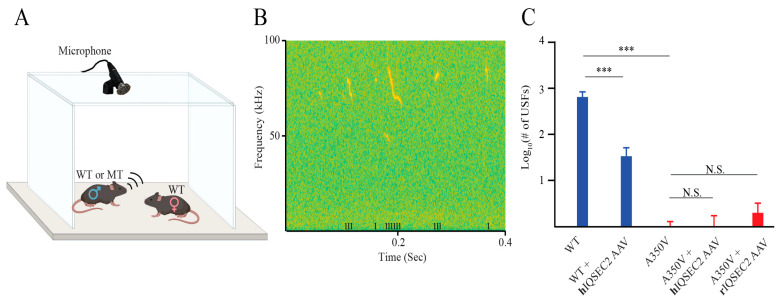
Neither the rIQSEC2 AAV nor the hIQSEC2 AAV rescues mating calls in A350V *Iqsec2* mutant mice. (**A**) Schematic of the methodology to measure ultrasonic vocalizations with a microphone. Wild-type (WT) *Iqsec2* or mutant (MT) A350V *Iqsec2* male mice were placed together with a WT female for five minutes. (**B**) A representative spectrogram showing the ultrasonic signals recorded during a 0.5 s segment of a 5 min free interaction of a WT male with a WT female. Ultrasonic fragments (USFs) identified by the TrackUSF system are marked at the bottom of the figure by the notation “I”. (**C**) Comparison of the number of USFs between WT and A350V mutant mice with and without treatment with the hIQSEC2 AAV and in A350V mice with the rIQSEC2 AAV. Treatment of WT mice with rIQSEC2 AAV was not performed. The Y-axis shows the Log10 transformation of the number of USFs (+1). Without treatment, A350V mutant mice emitted significantly fewer vocalizations than wild-type mice (Wilcoxon rank-sum test, *p* < 0.00001). Treatment did not affect the production of vocalizations of A350V mutant mice (Wilcoxon rank-sum test, *p* = 0.45 for hIQSEC2 AAV and *p* = 0.36 for rIQSEC2 AAV, *n* = 83 for A350V mutant mice with no treatment, *n* = 15 for A350V mutant mice with hIQSEC2, and *n* = 33 for A350V mutant mice with rIQSEC2 AAV), but the hIQSEC2 AAV did cause a significant reduction in vocalizations in WT male mice (Wilcoxon rank-sum test, *p* < 0.00001, *n* = 66 for WT mice with no treatment and *n* = 26 for WT mice with the hIQSEC2 AAV). *** *p* < 0.00001; N.S., not significant *p* > 0.05. Bar graphs indicate median values ± SE.

**Figure 6 ijms-26-08311-f006:**
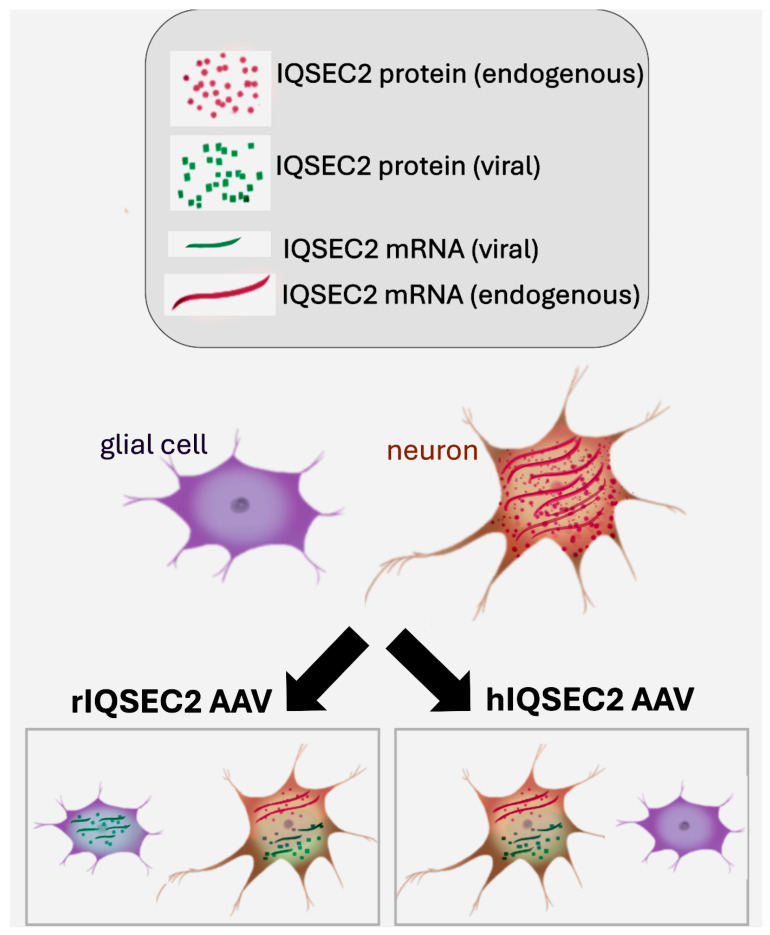
Proposed feedback inhibition mechanism for *IQSEC2* mRNA transcription. *IQSEC2* mRNA is normally only transcribed in neurons. Due to differences in the promoter specificity of the two viruses, the rIQSEC2 AAV results in an increase in rat *Iqsec2* mRNA and protein in all brain cells (neurons and glial cells), while the hIQSEC2 AAV results in an increase in human *IQSEC2* mRNA and protein only in neurons. IQSEC2 protein derived from the virus downregulates *Iqsec2* mRNA transcription from the endogenous mouse *Iqsec2* gene.

**Figure 7 ijms-26-08311-f007:**
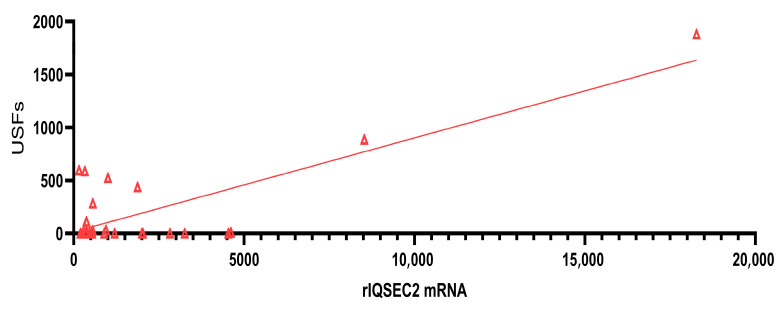
Correlation between rIQSEC2 AAV-derived rat *Iqsec2* mRNA and vocalizations in A350V male mice. There was a highly significant correlation between the expression of rat *Iqsec2* mRNA (fluorescent units) derived from the rIQSEC2 AAV determined by qRT-PCR and vocalizations in A350V IQSEC2 mice (*n* = 26; r = 0.80; *p* < 0.0001).

**Table 1 ijms-26-08311-t001:** Primers for qRT-PCR.

Gene	Primer Name	Description	Species	Direction	Sequence
*Iqsec2*	Endo-IQSEC2	Primers for measuring endogenous mouse mRNA in mice injected with hIQSEC2 AAV	Mouse	Forward	GCAAGAAACCGGTCCTGTCG
				Reverse	AAATTTTGGTGACCTCCCGC
*Iqsec2*	rAAV	Primers specific for viral IQSEC2 mRNA derived from rIQSEC2 AAV	Rat	Forward	CGGGTTTGCCGCCAG
				Reverse	GCCTCGGCTTTGTCGTCAT
*PGK1*		Housekeeping gene	Mouse	Forward	CACCGAGCCCATAGCTCCAT
				Reverse	CTGCAACTTTAGCGCCTCCC

## Data Availability

The original contributions presented in this study are included in the article/Appendix A. Further inquiries can be directed to the corresponding author.

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
