# Peer review of "Rescue of Iqsec2 Knockout Mice with Human IQSEC2 Adeno-Associated Virus Mediated Gene Therapy"

_ijms, 2025, doi:10.3390/ijms26178311_

Round 1
Reviewer 1 Report
Comments and Suggestions for Authors
The manuscript “Rescue of Iqsec2 knockout mice with human IQSEC2 adeno-associated virus mediated gene therapy” represents a short report comparing the ability of two different AAVs to rescue the phenotype caused by three different mutations in mouse Iqsec2 gene. While the therapeutic potential of developing AAV-mediated gene therapy for impaired Iqsec2 gene is strong, the experimental design and the depth of discussion are highly questionable.
The main critique is that the authors use two different AAVs – the old (previously published) rat Iqsec2 AAV, with a non-cell and non-tissue specific promoter and the novel human IQSEC2 AAV with neuron-specific promoter. While the promoters are briefly mentioned in the introduction, they are not emphasized in the results and not discussed in the discussion. On the contrary, the authors emphasize the origin of Iqsec2 sequences, calling the constructs “rat” vs “human”. Considering that the rat and human sequences are 98.9% identical (and 99.5% similar, based on pairwise protein alignment for the 1488 aa isoforms), these tiny sequence mismatches are highly unlikely to yield statistically significant differences in therapeutic ability of AAVs. (Moreover, if the authors wanted to emphasize the sequence origin, why not take the mouse sequence that is slightly more different from human and rat sequences (98.4% identical to human sequence) for the mouse experiments?) The difference in promoters, on the other hand, can make tremendous difference, as one is ubiquitous and the other is cell-specific. Additionally, they differ in length, which might be beneficial for AAV packaging capacity (it’s just 90 bp difference though, not much). Yet, the promoters are not discussed and the data confirming the expression pattern and level of expression of Iqsec2 under the two different promoters are missing. This severely undermines the significance and validity of the study.
Other significant limitations:
- In the intro, the authors emphasize the benefit of using shorter 4.9 kb insert compared to a longer 5.1 kb insert, however, such a small difference in size might not lead to noticeable change in the titer and quality of the virus. To claim this benefit, additional data need to be shown.
- Figure 3, B: The difference for the KO rescue (rat construct) for S1 and S2 values is very close to significance. Was the power analysis performed? Is there a reason why for the rat construct 13 mice were used while for the human rescue 16 mice were used? There might be a chance that significance would have been reached if the sample sizes were matching (16).
- It doesn’t make sense to try to further overexpress Iqsec2 in the A350V mice, as that mutation causes constitutively active GEF. These data can be a part of another follow-up study, following the study design that the authors outlined in the discussion. It doesn’t make sense to put these data here, as it is counter intuitive.
- The discussion is very short, with 1/3 of it discussing the experiment that doesn’t even make sense. It should focus on discussing the promoters, expression patterns, levels of expression in different AAVs, etc. Moreover, it is interesting to further dive into the harmful effect of IQSEC2 overexpression in WT animals. This is potentially a critical limitation to the translational capability of this treatment – a little too much and you are causing harm instead of good. This is a good place to talk about the (auto)regulation of IQSEC2 gene, is it easy to shift it’s expression level, what affects its expression in WT animals, how stable is overexpression using AAVs, etc.
Minor points:
Line 93 – please, rephrase “there was a significant interaction between Iqsec2 genotype and treatment on growth for both AAVs”. This sentence is hard to understand
Line 98 – “we found that there was no change in the growth rate with the rat IQSEC2 AAV”. In WT of KO?
Line 125 – “as compared to WT mice with an intermediate phenotype in heterozygous female mice” – please, rephrase. Hard to understand
Lines 134-140 are a part of Fig.2 legend. Please, change font and fix the line spaces
Line 182 – please, clarify that vocalization is normal. As it is written now it reads as if it is abnormal.
Line 184 – “we found that there was no benefit…” Not only there was no benefit. Looks like is was severely aggravated (40% vs 23% is a lot).
Reviewer 2 Report
Comments and Suggestions for Authors
The manuscript is well written and presents significant new data that would be of interest to the journal readers. The authors should address the following concerns/unanswered questions:
- There is no reasonable explanation as to why the rat transgene (unlike the human transgene) failed to improve the growth in hemizygous mice but had an adverse effect on heterozygous female KO mice. A more detailed explanation is necessary.
- There is no clear explanation for why, although the growth of the KO mice initially improved with the delivery of the human transgene, it slowed down after PND 21. A more detailed explanation is necessary.
- Is there a benefit of using a 4.9 kb AAV vector vs. a 5.1 kb vector in terms of the efficiency of the delivery? The previous study referenced in the manuscript (reference 11) shows similar results while using a 5.1 kb vector?
- Which splice isoform of the gene was used and why?
- Instead of double correction (shRNA+transgene), can the CRISPR-CAS9 approach be used in the gain-of-function mutations? This should be discussed.
- In human trials, overdose is an important possible adverse effect. What do the authors suggest as a potential way to decrease the risk of such overdosing?
- Minor issues:
Line 50: replace “in four cases that variants in the 50 IQSEC2 gene are associated with severe…” with “four affected individuals with IQSEC2 variants and severe…”
Line 57, replace “will likely require 24-hour access to an adult who can care for them for the rest of their lives” with “will need constant lifetime caregiver supervision”
Round 2
Reviewer 1 Report
Comments and Suggestions for Authors
Thank you for addressing the original concerns. The revised manuscript has been significantly improved, however, a few more limitations/comments need to be addressed before publication.
- The new data on the autoregulation of the Iqsec2 gene in A350V mutant mice are extremely valuable and provide new insights into the mechanism of the observed rescue. A discussion of the difference between the rescue potential of rIqsec2 and hIQSEC2 is adequate; however, it remains purely speculative unless protein expression is demonstrated. For example, some small tag (such as FLAG) can be linked to the viral IQSEC2 to distinguish between endogenous vs viral protein by immunostaining.
- Figure 4 legend, as well as description of the results, state that hIQSEQ can rescue the social behavior of the mutant mice. Even though the difference is statistically significant, it is far from the WT values, so the word “rescue” should be changed to “partial rescue.”
- Lines 208-211(rIQSEC2 AAV rescue of A350V male vocalization) – why aren’t these data shown in Fig. 5C?
- Discussion on introducing miRNA binding sites (lines 330-338) is great; however, one more thing needs to be discussed here. Introducing any kind of 3’UTR would increase the size of the viral genome, and this is exactly what you were trying to avoid (and showed that even a small shift from 5.1 to 4.9kb size matters). This limitation needs to be mentioned. Moreover, 3’UTRs, unlike CDSs, are much less evolutionary conserved, and even though the miRNA binding sites may be conserved (sequence-wise), their functionality might not. It really depends on the dynamics of miRNA expression in WT vs mutant mice or humans, the "strength" of the binding sites, etc. The translational value of mouse studies might be compromised. It is not necessary to mention this part in the discussion though; it is just something to keep in mind if you ever decide to pursue this study.
Minor points:
- It would be good to change the subheadings in the results section: state the exact name of the mutations instead of “model#1 at Columbia University” and “model#2 at Shinshu University”. Also, please, introduce the principle of each mutation at the beginning of the corresponding sections: around lines 106 and 165. You can copy-paste or rephrase the descriptions from method section.
- Figure 5C – it’s better to change MT on the x-axis to A350V, for consistency.
Thank you!
Reviewer 2 Report
Comments and Suggestions for Authors
The authors made the requested changes and addressed my concerns. The manuscript is improved and may be considered for publication.
